# Therapeutic Targeting of Glioblastoma and the Interactions with Its Microenvironment

**DOI:** 10.3390/cancers15245790

**Published:** 2023-12-10

**Authors:** Vassilis Genoud, Ben Kinnersley, Nicholas F. Brown, Diego Ottaviani, Paul Mulholland

**Affiliations:** 1Glioblastoma Research Group, University College London, London WC1E 6DD, UKb.kinnersley@ucl.ac.uk (B.K.);; 2Department of Oncology, University College London Hospitals, London NW1 2PB, UK; 3Department of Oncology, University Hospitals of Geneva, 1205 Geneva, Switzerland; 4Centre for Translational Research in Onco-Haematology, University of Geneva, 1205 Geneva, Switzerland; 5Guy’s Cancer, Guy’s & St Thomas’ NHS Foundation Trust, London SE1 3SS, UK

**Keywords:** glioblastoma (GBM), heterogeneity, tumour microenvironment (TME), myeloid cells, microglia, tumour associated macrophages (TAMs), therapeutic targeting

## Abstract

**Simple Summary:**

Glioblastoma (GBM) is the most aggressive brain tumour. Patients with GBM have a dismal survival and there is a distinct lack of curative treatments. We are increasingly understanding that the GBM tumour is composed not only of tumour cells but a complex tumour microenvironment (TME) of neuronal, glial and immune cells. Research into this area is important because the diversity of tumour cells and interactions with the TME contribute to the aggressiveness and treatment resistance of GBM. In this work, we review the multiple types of cells forming GBM and their interactions and provide examples of how our improved understanding can suggest potential new treatment strategies for this devastating disease.

**Abstract:**

Glioblastoma (GBM) is the most common primary malignant brain tumour, and it confers a dismal prognosis despite intensive multimodal treatments. Whilst historically, research has focussed on the evolution of GBM tumour cells themselves, there is growing recognition of the importance of studying the tumour microenvironment (TME). Improved characterisation of the interaction between GBM cells and the TME has led to a better understanding of therapeutic resistance and the identification of potential targets to block these escape mechanisms. This review describes the network of cells within the TME and proposes treatment strategies for simultaneously targeting GBM cells, the surrounding immune cells, and the crosstalk between them.

## 1. Introduction

Glioblastoma (GBM) is a disease of significant unmet need with a median overall survival of only nine months [1]. The standard treatment for GBM comprises maximal safe surgery, adjuvant radiotherapy (RT) with concurrent temozolomide (TMZ), and then adjuvant TMZ [2]. Despite there being many clinical trials over the last two decades, the management of the disease has not changed. Methylation of the O6-methylguanine-DNA methyltransferase (MGMT) promoter is the most impactful prognostic biomarker, and it confers a better prognosis and predicts response to treatment [3].

Although novel treatments have not demonstrated substantial improvement in overall survival (OS) in phase III clinical trials, subsets of patients with GBM appear to benefit. Bevacizumab, an anti-vascular endothelial growth factor (VEGF) monoclonal antibody (Ab), was shown to improve progression-free survival (PFS) but not OS [4]. Tumour-treating fields (TTFs), an approach that uses alternating electrical fields, were reported to confer a modest OS benefit in newly diagnosed patients with GBM [5] but has not been widely adopted. Recently, the therapeutic vaccine DC-VaxL, which adds dendritic cell (DC) therapy to standard therapy in newly diagnosed patients, demonstrated a median OS of 19.3 months compared to 16.5 months in matched historical controls [6]. Immune checkpoint inhibitors (ICIs) with anti-PD-1 and anti-CTLA4 have radically changed the treatment of many other tumour types but have not demonstrated a survival advantage in GBM thus far [7,8]; however, further evaluation is ongoing, with neoadjuvant ICI showing promising initial results [9,10]. Finally, the remarkable success of chimeric antigen receptor (CAR) T-cell therapy in haematological cancers drives the exploration of this therapy in solid tumours, including GBM [11].

Clinical trials have demonstrated that GBM is largely resistant to immunotherapeutic approaches as it is an immunologically “cold” tumour [12]. The low immunogenicity of GBM is characterised by low T cell infiltration and absence of DC, a low tumour mutational burden, limited MHC-I expression, a highly immunosuppressive tumour microenvironment (TME), and a lack of conventional draining lymph nodes [13,14,15]. More detailed analyses of how treatments impact GBM and its TME are needed to identify potential therapeutic limitations and resistance pathways. Additionally, studies focusing on the origin of GBM and the complex crosstalk between a tumour and the surrounding cells will further improve our understanding of tumour development and the dynamic response to treatment. Additionally, detailed molecular analyses have aided the understanding of the cellular interactions and transcriptional pathways underlying GBM tumour composition. Identifying shared features between injuries and tumorigenesis has helped us understand the impact of inflammation on brain parenchyma and its potential link to the origins of GBM [16].

In this review, we will first describe our current understanding of the origins and genetic evolution of GBM. We will then focus on the interplay among tumour cells, astrocytes, and neurons as a cell network. The role of infiltrating immune cells in GBM will be described, and we will conclude with the therapeutic opportunities offered by targeting these cells and their interactions.

## 2. Molecular Subtype and GBM Stem Cells

In 2010, as part of the TCGA project, a pivotal paper by Verhaak et al. described four distinct molecular subtypes of GBM: proneural, neural, classical and mesenchymal [17]. Key genetic driver alterations are associated with each subtype: *PDGFRA* amplification for proneural, *EGFR* overexpression or amplification for neural, *EGFR* mutation, amplification, or overexpression for classical, and *NF1* mutation or deletion for mesenchymal [18]. Subsequently, as molecular techniques evolved from bulk to single-cell tumour profiling, similarities with the normal neural lineage were identified [19]. These lineages are referred to as oligodendrocyte-progenitor-cell-like (OPC-like), neural-progenitor-cell-like (NPC-like), and astrocyte-like (AC-like). Additionally, mesenchymal-like (MES-like) cells found in GBM are not observed in normal neural development but are induced in response to injury [19].

The cell of origin of GBM is still debatable. Evidence points towards either neural progenitor cells (NPCs) or oligodendrocyte progenitor cells (OPCs), as experimentally inducing somatic mutations in stem cells from either NPCs or OPCs can lead to tumour development [20]. Additionally, the subventricular zone (SVZ) has been identified as a potential stem-cell niche and the possible site of origin of GBM [21]. While the cell-state composition of the tumour mass is likely to be impacted by the cell of origin, its evolution is also influenced by various parameters, including mutations acquired during tumour development and interactions with the TME [22,23,24]. Specifically, *TP53* or *NF1* mutations in neural stem cells (NSCs) lead to the development of OPC progeny [21]. There is also high intra-tumoural heterogeneity, with the coexistence of up to five spatially resolved gene signatures, including NPC-like, OPC-like reactive astrocytes, MES-like, and MES-hypoxia, in the majority of GBM tumour samples analysed in a recent study [25].

As a tumour grows, it induces injury-associated inflammation, disrupting the physiological organisation of the brain parenchyma via mechanical, chemical, and even surgical or treatment-related impacts [26]. Injury-associated inflammation and wound response gene expression signatures have been described, and they can account for a significant proportion of the transcriptional heterogeneity of GBM [27]. Furthermore, the presence of GBM stem cells (GSCs) in the tumour bulk is positively correlated with tumour invasion, and OPCs can migrate to injury sites and proliferate to differentiate and promote remyelination [28].

Specific GBM molecular subtypes can impact prognosis, with the MES-like signature and the clustering pattern of AC-like tumour cells being associated with poor patient outcomes [29]. However, the prognosis is improved if the AC-like tumour cells are dispersed and in contact with other subtypes [30]. The MES-like subtype displays features of reactive astrocytes and is induced by the presence of tumour-associated macrophages (TAMs), which are enriched in regions of the tumour with the MES-like signature [31]. RNA velocity analysis suggests that MES-like cells are derived from an AC-like state in most tumours [30], but direct evolution from NPCs to MESs has also been described, drawing into question the necessity of the development of a transient astrocyte state [32].

Overall, the described molecular signatures illustrate the complexity of cellular interactions and dynamics within GBM tumours, with GSC plasticity driving evolution to escape treatment effects and leading to resistance to therapy and tumour recurrence [33,34]. Understanding the evolution and dynamics of these molecular subtypes will improve our understanding of the impact of treatment.

## 3. Blood–Brain Barrier Disruption in GBM

The physiological blood–brain barrier (BBB) is composed of astrocyte endfeet and tight junctions between endothelial cells and pericytes, which limit the accessibility of pathogens and many molecules, including chemotherapy, to the central nervous system (CNS) [35]. GBM growth induces physical distortion of the BBB, and the associated inflammation with neo-angiogenesis leads to the formation of leaky blood vessels, further disrupting its integrity [36,37]. Consequently, tumours are more permeable and show an increase in blood perfusion. VEGF is the major cytokine driving neo-angiogenesis; it decreases the expression of intercellular adhesion molecule (ICAM-1) and vascular cell adhesion molecule 1 (VCAM-1), thus limiting the adhesion and infiltration of immune cells to and into the tumour [38]. The aberrant and disorganised vascular morphology within the GBM tumour also limits the distribution of oxygen or nutrient supplies, leading to hypoxia and treatment resistance [39,40,41]. Through the induction of the transcription factor HIF1⍺, hypoxia regulates many genes implicated in diverse features, such as angiogenesis, survival, treatment resistance, genomic instability, and invasion [42,43,44]. Expression of HIF1⍺ is associated with an increase in MDR1, which can drive chemoresistance. In vitro experiments in GBM cells have demonstrated hypoxia-induced resistance to cisplatin, TMZ, and etoposide, as well as the overexpression of CD133, a marker associated with stemness [42,45,46]. As hypoxia limits the generation of reactive oxygen species (ROS), it can dampen the effectiveness of RT [46,47], and the limited perfusion of the tumour core causes necrotic areas and hypoxic regions that attract macrophages with immunosuppressive functions to develop [48].

## 4. The Complex Glioblastoma Microenvironment

The CNS parenchyma comprises neurons, astrocytes, oligodendrocytes, and microglial cells with complementary functions. Improved understanding of the complex crosstalk between the highly heterogeneous GBM cells and diversity of cell types in the surrounding TME will hopefully elucidate novel antitumour therapeutic strategies.

### 4.1. Astrocytes

Astrocytes maintain the structure and balance of the brain’s parenchyma as they buffer the metabolic environment and provide an energy substrate (glucose and lactate) for neurons [49]. In the presence of GBM tumour cells, astrocytes detect damage-associated molecular patterns through pattern recognition receptors and sense mechanical stress and injury. These triggers activate a specific molecular signalling network through the JAK/STAT pathway, leading to reactive astrocytes in a process known as astrogliosis [50]. Along with JAK/STAT activation, reactive astrocytes overexpress GFAP and can express the immunosuppressive molecule PD-L1 [51,52]. Reactive astrocytes produce high levels of immunosuppressive chemokines in the TME, including IL-10 and TGF-β [51]. As a feedback loop, IL-10 and IFNγ induce the reactive state of astrocytes through JAK/STAT activation [51,53]. Increased extracellular ATP from parenchymal damage leads to ATP secretion from astrocytes and the recruitment of microglia. Moreover, pro-inflammatory reactive astrocytes are less efficient at maintaining the integrity of the BBB and further contribute to local inflammation [51,54,55]. Thus, given their role in inducing the immunosuppressive state of the GBM TME, limiting the formation of reactive astrocytes could favour therapeutic interventions.

### 4.2. Neurons

GBM tumours comprise a complex neuronal and axonal architecture [56], and they can directly form synapses with glioma cells to drive their proliferation via glutamatergic signalling [57]. GBM cells also interact with neurons through paracrine signalling with neuroligin-3, brain-derived neurotrophic factor, and AMPAR-mediated excitatory electrochemical synapses from neurons, promoting tumour growth [58,59,60,61]. Similarly, GBM cells can influence neurons through the secretion of non-synaptic glutamate [62,63] and can reduce the activation of inhibitory interneurons [64]. Gliomas may also increase the risk of epilepsy by influencing glutamatergic and GABAergic signalling in neurons [65], with studies in awake patients observing more neuronal excitability in the GBM-infiltrated cortex [59]. Short-range electrocorticography on the tumour-infiltrated cortex revealed functional remodelling of language circuits as some tumour regions with TSP-1^+^ tumour cells maintained functional connectivity with neurons. This molecularly distinct GBM subpopulation is responsive to neuronal signals and has a synaptogenic, proliferative, invasive, and integrative profile, ultimately conveying a poor prognosis [16]. Moreover, it has been proposed that gliomas originating from functionally connected cortical regions are more connected to neurons and will promote the invasion of specific GBM subpopulations [16].

### 4.3. Glioblastoma Connectome

Large subpopulations of GBM cells are connected via tumour microtubes (TMs), which are the mechanical base for a tumour cell network, through gap junctions [66,67]. TMs can support the exchange of cell nuclei, microvesicles, mitochondria, Ca^2+^, and chemotherapy molecules and are ultimately associated with treatment resistance [66,67]. Glutamatergic synapses in these networks can activate tumour cells and are associated with increased tumour cell proliferation and invasion [58,59,60,61]. These connections between tumour cells have been linked to invasiveness in other malignancies, such as breast, colon, and prostate cancer [68], and more recently to GBM [69]. Historically, it was believed that GBM invades the surrounding brain parenchyma by following anatomic structures, such as blood vessels, nerves, and astrocytic tracts [70]. However, a recent study identified GBM tumour cells with a neuronal molecular state that lacked connections with other tumour cells or astrocytes and that were the main drivers of diffuse brain invasion [16,71]. The molecular state of unconnected tumour cells was more enriched for neuronal and NPC-like cells and less enriched for MES-like cells [16]. GBM cell invasion resembles neuronal migration during development, with glutamatergic stimulation of synapses increasing the invasiveness of GBM cells, leading to TM formation. After invading surrounding tissue, invading unconnected tumour cells connect with the tumour mass and surrounding astrocytes to form what has been described as the “GBM connectome” [16]. Connected tumour cells are enriched for astrocytic, mesenchymal, and non-neuronal cell states that are consistent with injury response states. Confirming these findings, the upregulation of neuronal signalling programs is associated with invasiveness at recurrence of GBM [72].

### 4.4. Myeloid Cells

Myeloid cells in GBM comprise bone-marrow-derived macrophages and microglia, which are brain-resident cells originating embryologically from the yolk sac. These two populations represent more than 95% of the immune infiltrate in GBM [73]. In the healthy brain, microglia secrete neurotrophic factors and promote synaptic pruning to maintain CNS homeostasis [74,75]. Gliomas with an increased number of tumour-associated macrophages (TAMs) are associated with a higher grade and poor survival at diagnosis and at recurrence [76,77,78].

In GBM, TAMs are predominantly pro-inflammatory and secrete various cytokines, including TNF⍺, IL-1a, IL-6, and IL-12, but they also have immuno-suppressive properties, which are mainly through IL-10 and TGF-β secretion [79,80]. TAMs recruit polymorphonuclear neutrophils (PMNs) that generate ROS and nitric oxide (NO), further contributing to the inflamed and immunosuppressive TME. Single-cell RNA sequencing (scRNA-Seq) studies have described an enriched TAM population in the core of the tumour, which harbours a pro-inflammatory phenotype, whereas, at the tumour periphery, microglia with anti-inflammatory properties are more prominent [81,82,83,84]. Differences in the myeloid component of GBM were also noted between the tumour at diagnosis and at recurrence, as proportionally more microglia cells were present in GBM at diagnosis and proportionally more TAMs were present in recurrent GBM [85]. TAMs promote the differentiation of GBM towards the MES-like signature mainly through TNF⍺, C1q, IL-1a, and IFNγ [32], and the loss of *NF1* and *PTEN* in MES-like GBM cells is associated with greater TAM infiltration [86].

TAMs are highly plastic and dynamic, with their different states being defined by multiple gene signatures [85]. M1 and M2 phenotypes were proposed to distinguish classically and alternatively activated macrophages, respectively [87]. In vitro, M1 macrophages are induced after exposure to pro-inflammatory cytokines (such as TNF⍺ or IFNγ) and produce pro-inflammatory factors [88]. M2 differentiation is triggered by anti-inflammatory cytokines (such as IL-4, IL-10, and IL-13), and these macrophages are less cytotoxic and produce immunosuppressive cytokines, such as IL-10 and TGF-β [89]. In GBM, most TAMs are described as being polarised to M2 and will limit T cell activity, secrete extracellular matrix components, and stimulate angiogenesis [43]. Arguably, this dichotomy is mainly observed in vitro, as in GBM patients, scRNA-Seq could not confirm the presence of these two polarised M1/M2 stages [90,91]. Although CD163 and CD206 have been proposed to help discriminate between the two extremes [91], many macrophages co-express M1 and M2 markers, and instead, a continuum with many states between the M1 and M2 spectrum exists [92,93]. Moreover, dynamic changes in the tumour and the TME that are driven by tumour growth and therapeutic interaction impact the TAM phenotype [88]. With single-cell resolution enabling the detection of smaller populations that can be diluted in bulk analysis, new TAM subtypes in the GBM TME have recently been described. The MARCO^hi^ macrophages and CD163^+^-HMOX1^+^ microglia are present only in mesenchymal GBM [94,95], with MARCO^hi^ macrophages inducing the mesenchymal transition and with HMOX1^+^ microglia driving T cell exhaustion. High-grade glioma-associated microglia are proliferative and proinflammatory, and they promote GBM progression through the induction of the inflammasome [96]. CD73^hi^ macrophages are immunosuppressive, and their signature persists after anti-PD-1 treatment [97]. This population does not directly impact prognosis, but knocking out CD73 in mice led to increased iNOS^+^ myeloid cells and enhanced antitumor efficacy of an ICI treatment [97], indicating that CD73^hi^ macrophages drive some immunosuppressive TME features.

Myeloid-derived suppressor cells (MDSCs) represent another myeloid population in the TME and are divided into monocytic and granulocytic MDSCs [98]. MDSCs harbour immunosuppressive features as they increase the catabolism of L-arginine by using arginase-1, depleting an essential amino acid for T cell proliferation. MDSCs can also generate ROS, impacting T cell efficacy [99], and they contribute to IL-10 and TGF-β secretion in the TME.

Neutrophil infiltration is associated with poor prognosis and treatment resistance in GBM [100]. A recent study identified a substantial neutrophil infiltration in the GBM TME [101], with CXCL8 and IL-8 being the main chemokines attracting neutrophils to the centre of the tumour [102]. Neutrophils secrete elastase, promoting tumour proliferation and angiogenesis [103], and they contribute to the immunosuppressive TME through the secretion of arginase-1, GM-CSF, and S100A4 [100,104].

### 4.5. Dendritic Cells

DCs are key to bridging innate and adaptive immunity by presenting antigens (Ag) to T and B cells [105]. They are not seen in the physiological brain parenchyma but, rather, during chronic inflammation. CCL5 and XCL1 can recruit DCs to the GBM TME, as observed in mouse models [106]. Based on the scRNA-Seq profiling of GBM tumours, it was estimated that about 4.5% of cells in the GBM TME have a molecular DC signature [73]. If present, DCs can secrete IL-12, which recruits CD8^+^ T cells and reinvigorates anergic T cells [105]. Regulatory DCs have also been described and can promote regulatory T cells (Tregs) [107] while limiting CD8^+^ T cell recruitment [108].

### 4.6. Lymphoid Cells

Natural killer (NK) cells exert contact-dependent cytotoxic activity through the secretion of granzyme B and perforins [109]. NK cells are found in the GBM TME [110], but radio-chemotherapy can decrease their presence [111]. NK cells are inhibited by the expression of MHC-I, which is often downregulated in GBM [112], and NK cell functionality is enhanced when NKp30 binds to B7-H6 on tumour cells [113].

T cells are the hallmark effectors of anti-tumour immunity. Their cytotoxic activity can lead to tumour eradication in other cancers and in mouse models of GBM, especially with ICI treatment [114]. However, generally, only a small number of T cells are present in the GBM TME, and when present, they exhibit an exhausted phenotype due to chronic exposure to Ag, which is associated with a lack of stimulation that leads to ineffective anti-tumour activity [115]. Many studies have explored the potential of ICI to enhance T cell activity in patients with GBM, but these have mostly failed to demonstrate significant clinical improvement [7]. Two companion studies have, however, observed that the neoadjuvant administration of anti-PD-1 Ab showed a signal towards improved OS, which will need to be validated in larger cohorts [9,10].

Tregs are immunosuppressive T cells, and their presence in GBM is associated with poor prognosis [116,117]. In contrast to CD4+/CD8+ T cells, Tregs highly express the transcription factor FOXP3, which downregulates the expression of pro-inflammatory cytokines such as IL-2 through the induction of two other transcription factors, NFAT and NFkB [118]. Innate immune cells secrete CXCL9, -10, -11, CXCR3, and CCL5-CCR5, which attract Tregs in the GBM TME [119]. As an autocrine loop, Tregs also secrete IL-10 and TGF-β, which further promotes the transition of T cells into Tregs and supports local immunosuppression [120,121]. Tregs also express high levels of the immunosuppressive molecules CTLA-4, PD-1, and GITR, thus providing inhibitory signals for all infiltrating immune cells.

## 5. Therapeutic Perspectives

Understanding the complex interactions between the multiple cell types underlying the GBM TME and their respective states or molecular profiles can help in the design of treatment strategies. Figure 1 illustrates these different cell types, and Table 1 highlights potential therapeutic approaches targeting the different components of GBM, as well as potential anticipated limitations.

No new systemic anti-cancer therapies have been approved for GBM for nearly twenty years, and whilst many cancers have benefited from ICI, GBM has been left trailing behind. Clinical trials evaluating ICIs have had little or no impact on the overall clinical outcomes of patients with GBM. However, case reports have demonstrated the potential for clinical benefit in an as-yet unidentified subpopulation of GBM, and early trials of neo-adjuvant PD-1 therapy appear promising, as do other forms of immunotherapy, such as CAR-T cell therapies and the recent results from DC Vax-L [6,122,123,124,125]. Improved outcomes from these immunotherapies were mainly observed in patients with methylated MGMT promoter tumours. This raises the question of these tumours having a potentially more immunotherapy-permissive TME and warrants further investigation.

**Table 1 cancers-15-05790-t001:** Examples of therapeutic strategies for targeting GBM and TME interactions. ICI, immune checkpoint inhibitor; TMB, tumour mutational burden; Ab, antibody; TAMs, tumour-associated macrophages; DCs, dendritic cells; NK, natural killer; BBB, blood–brain barrier; TME, tumour microenvironment; GBM, glioblastoma; GSCs, glioblastoma stem cells; DCs, dendritic cells.

Cell Targeted	Mechanism Targeted	Potential Strategy	Potential Limitations
*GBM*			
Tumour cells	Enhancing immunogenicity	Increasing MHC-I expression Increasing TMB and, therefore, neoantigen presentation with RT/TMZ treatment	Dampened NK cell response Sub-clonal TMB is associated with poor response to ICIs Effect may be restricted to MGMT-methylated GBM
Tumour cells, immune cells	Blocking negative regulators of antitumour immune response	ICIs	Lack of T cell infiltration, highly immunosuppressive TME
GSCs	Targeting specific markers Promoting GSC differentiation	Inhibition of CD133/ GPD1/ L1CAM Graphene oxide, Sulindac	Lack of truly specific targets, intrinsic treatment resistance
Tumour cells	Limiting the impact of hypoxia	HIF1⍺ inhibition	Limited efficacy thus far
*TME—normal brain*			
Astrocytes	Limiting astrogliosis to suppress reactive astrocyte formation	JAK/STAT inhibition	Limited data available
Neurons	Targeting AMPAR signal	Perempanel treatment	Limited data available
Connectome	Targeting gap junctions	Connexin 43 targeting	Limited data available
*TME—immune component*			
TAMs	Limiting infiltration	CCL2 inhibition	Limited data available
	Limiting M2 polarization	CSF-1R inhibition	Induced resistance
	Enhancing phagocytosis	CD47 inhibition	Haematological side effects
	Depleting specific populations	CD73^+^ or MARCO^high^ depletion	Limited data available
DCs	Enhancing immune activation	Therapeutic vaccines	Efficacy dependent on T cell homing to the tumour
Neutrophils	Limiting infiltration	Blocking chemokines	
Tregs	Depleting cells	IL-25 depleting Ab	Limited data available
T cells	Enhancing targeting and activation	CAR-T cell therapy	Antigen loss, on-target off-tumour effect
NK cells	Enhancing activation	Activating cytokines	Limited data available
*CNS—integrity*			
BBB	Increasing permeability	Focused ultrasound	Transient effect

**Figure 1 cancers-15-05790-f001:**
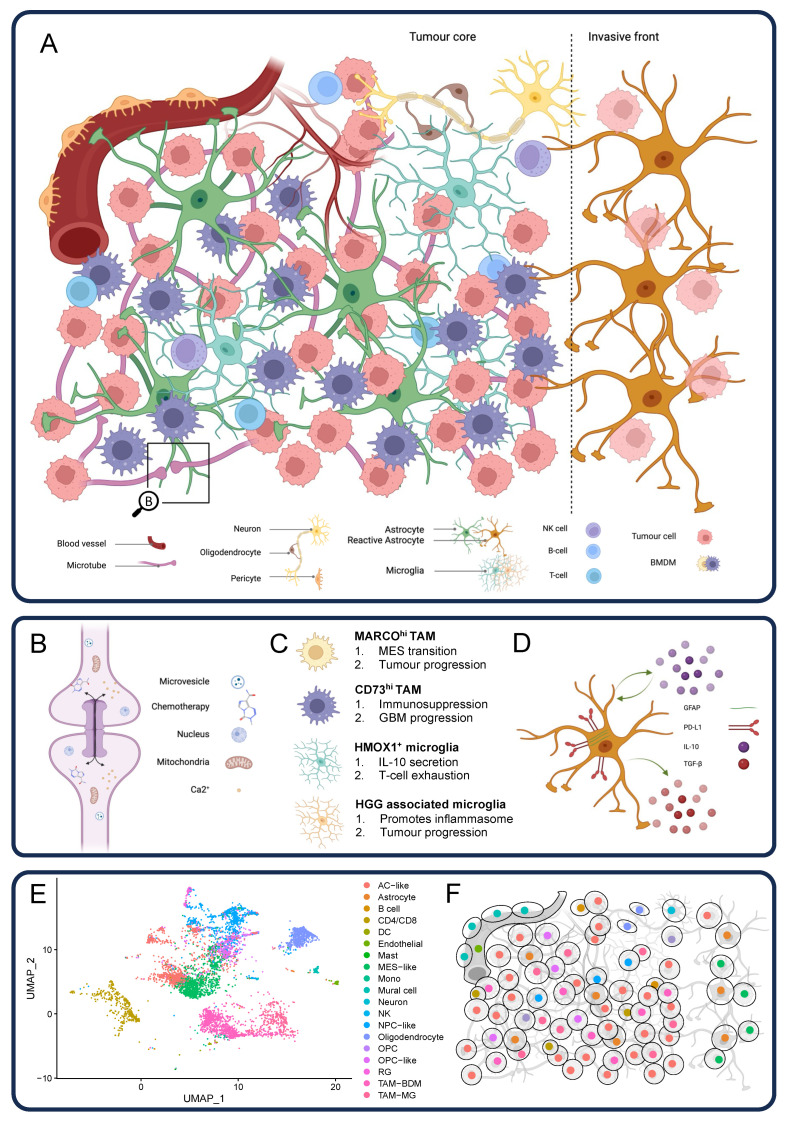
**The complexity of the glioblastoma tumour microenvironment and the differences in techniques used to analyse it.** (**A**) Schematic representation of the multiple cells composing the microenvironment of GBM. (**B**) Detailed representation of tumour microtubes and the different elements exchanged through this system. (**C**) Illustration of recently identified subpopulations of myeloid cells (MARCO^hi^ tumour-associated macrophages (TAM), CD73^hi^ TAM, HMOX1^+^ microglia, and high-grade glioma (HGG)-associated microglia) and their associated primary biological functions. (**D**) Representation of reactive astrocytes expressing GFAP and PD-L1 and secreting the immunosuppressive cytokines IL-10 and TGF-β. (**E**) UMAP plot indicating the diversity of cell populations in GBM. Single-nucleus RNA-Seq data were obtained from GBM-CPTAC [126], processed using Seurat [127], and mapped to GBmap reference cell types [128] using Azimuth [129]. (**F**) Illustrative representation of spatial transcriptomics with single-cell resolution offered by new technological advances. This allows the identification of the complexity of the GBM TME by using gene expression to identify the different cells while keeping the biological architecture. The figure was created with BioRender.com, accessed on 16 October 2023.

Specific glioblastoma cell populations play prominent roles in tumour resistance or invasion. However, targeting these populations may not be sufficient as many other players support tumour growth in the TME. While the genomic profiling of GBM has identified several key dysregulated signalling pathways [130], targeted therapies have been ineffective in GBM, with intra-tumoural heterogeneity being a likely key factor driving this treatment resistance. The interaction between GBM cells and the TME drives immunosuppression and tumour progression. Consequently, an effective approach to therapy in glioblastoma is likely to involve simultaneously targeting GBM cells, the surrounding immune cells, and the crosstalk between them.

Multipotent GSCs drive tumour initiation and self-renewal, which supports tumour progression and therapy resistance. Therapeutic strategies targeting GSCs could increase treatment success [42,43,47,131,132]. In vitro studies showed that while TMZ initially inhibits GSC proliferation, MES-like tumour cells are resistant to this treatment, driving tumour recurrence [133,134]. The altered metabolic pathways of GSCs have been successfully targeted with epigallocatechin gallate, a bioactive polyphenol inhibiting transglutaminase, which has been shown to restore the sensitivity of GSCs to TMZ and inhibit their proliferation in vitro [135]. However, it remains unclear if this strategy will translate to clinical trials. Additionally, targeting newer GSC-specific markers, such as the stem-like cell marker SOX2 that promotes tumour progression [136] or *SOCS3*, *USP8*, and *DOT1L,* which were recently linked to GSC growth, may be more efficient [137].

The molecular subtype of GBM cells is dynamic during treatment, as GBM cells with NPC- and OPC-like tumour signatures, which are more sensitive to RT, shift to an MES-like signature after RT. Consequently, RT enriches tumours for the MES-like subtype, which is more resistant to RT, which is possibly because of their proximity to RT-resistant reactive astrocytes. Additionally, RT leads to an acute accumulation of TAMs in the peritumoral area, driving resistance to RT, as TAM depletion restores sensitivity to RT [138,139]. Altogether, these changes in the TME after RT lead to an overall MES-like transformation, which is also the most stable subtype throughout various treatments [140].

Reactive astrocytes resist apoptosis through Fas or TRAIL pathways and have more DNA damage repair pathways than normal astrocytes [141]. Reactive astrocytes induced by activating injury pathways at the tumour margin can cause recurrence in that location [142]. This is not only due to tumour cells remaining after treatment but also because of their pro-tumoral profile [142]. Limiting the reactive injury response would, therefore, be a rational strategy for limiting tumour growth and resistance. The induction of reactive astrocytes mainly depends on the activation of the JAK/STAT pathway, which could be limited by specific inhibitors such as ruxolitinib [51]. Another strategy could be to target the tumour–astrocyte connectome, as this can drive treatment resistance. The inhibition of connexin 43 in TMs showed promising results in a pre-clinical model when combined with TMZ [143]. Additionally, in vitro experiments targeting AMPAR signalling with the anti-seizure medication perampanel limited the neuronal input to tumour cells and showed a synergistic effect with TMZ [144].

TAMs are critical for promoting tumour progression and immunosuppression in GBM and have likely contributed to the failure of immunotherapies in GBM thus far. Limiting TAMs’ infiltration or modulating their pro-tumoural polarisation are promising therapeutic strategies. The CCL2/CCR2 chemokine axis is implicated in TAM infiltration, and the inhibition of CCL2 led to lower TAM recruitment and improved ICI efficacy in a mouse model of GBM [145,146]. However, the efficacy may have been blunted by the increased immunosuppressive microglia observed in CCR2 knockout mice [85]. TAMs express CSF-1R, which regulates survival and key TAM functions. The modulation of CSF-1R signalling can be used to reprogram TAMs and limit immunosuppressive polarisation. The small molecule BLZ945 inhibits CSF-1R. Although it does not deplete TAMs, it induces polarisation towards a more M1 phenotype, reducing tumour progression [138]. The concurrent inhibition of the IGFR/PI3K pathway should be explored, as GBM cells can acquire resistance to BLZ945 through this pathway [147]. BLZ945 also enhances the initial response to RT [139] and the efficacy of anti-PD-1 therapy [148]. However, CSF-1R inhibition with PLX3397 monotherapy in humans was ineffective [149]. The CD47/signal-regulatory protein alpha (SIRPα) pathway is a critical innate macrophage checkpoint, as CD47 expression on tumour cells limits the phagocytic function of TAMs. [150]. In mice, depleting CD47 increases the phagocytosis of macrophages and limits glioblastoma tumour progression [151]. Targeting specific TAM subpopulations, such as CD73^+^ cells, extended the survival of mice with glioblastoma and showed synergistic effects when combined with anti-PD-1 and anti-CTLA-4 therapy [97]. In the same manner, Ab targeting of MARCO^hi^ macrophages limited the transition of tumour cells to the MES-like subtype and the stemness features of GSCs [95].

Considering the failure of ICIs in GBM, accumulating evidence points towards the lack of efficient T cell priming by DCs [152]. A recent study compared the immune profiles of brain metastases with those of primary GBM tumours. As the primary tumours were not in the CNS, metastases exhibited more efficient T cell priming by peripheral DCs, generating a more effective T cell response against brain metastases compared to that of GBM [153]. FLT3L-mediated DC population expansion led to enhanced immune priming in a mouse model of GBM [154], and the recent encouraging results of DC-Vax-L [6] support the exploration of DC-targeted therapies.

CAR-T cell therapy for GBM could bypass the need for local T cell activation, as lymphocytes are activated ex vivo. CAR constructs targeting single tumour antigens have shown only occasional clinical improvements, but the field is now moving toward targeting multiple antigens with multivalent CARs, which is supported by a pre-clinical study where a trivalent CAR directed toward HER2, IL13Ra2, and EphA2 showed better cytotoxicity than that of monovalent CAR T cells [155].

Depleting Tregs by targeting CD25 initially faced limitations, as the anti-CD25 antibody also blocked bystander IL-2 receptor signalling, limiting T cell antitumour activity [156]. However, recent developments showed that improved antibody specificity could efficiently deplete Tregs without impacting IL-2 signalling, supporting further evaluation of this strategy [156].

Targeting other immune cells could help shape a more immuno-permissive TME. The administration of NK cells activated with IFNγ, IL-2, and anti-CD3 to enhance their cytotoxicity improved PFS in a phase III clinical trial, although the primary endpoint of improved OS was not met [157].

Finally, as the BBB and irregular neovessels limit the optimal delivery [158] and therapeutic concentrations of drugs [159], the recent development of transient BBB opening with focused ultrasound [160] could pave the way for new chemotherapeutic protocols or large molecules, such as antibody or antibody–drug conjugates, in patients with GBM.

## 6. Conclusions and Future Directions

Advances in genomic techniques have led to a better understanding of the intratumoural heterogeneity of GBM, although this has yet to make a therapeutic impact. With the development of spatial transcriptomics, our understanding of the interactions within the TME will be further improved and may be used to identify location-dependent cell interactions that are lost in single-cell data (Figure 1), and this technology is already improving with the development of sub-cellular resolution. For example, Figure 1E depicts the diversity of cells composing human GBM based on a single-cell RNA sequencing analysis. Cells are clustered based on the similarities in their gene expression profiles. Figure 1F illustrates a spatial transcriptomics readout in which the histological cell architecture is maintained, and single-cell identities are similarly determined according to their gene expression profiles.

Identifying the different components of the TME, their physical connections, and their interactions reveals new mechanisms of targeting glioblastoma. As illustrated by previous clinical studies, targeting only one subpopulation of GBM tumour cells or only one pathway, such as EGFRvIII^+^ or PD-1, is not effective due to target loss and tumour resistance. Concurrent targeting of tumour cells and the TAM compartment or reactive astrocytes has therapeutic potential, as they form a connectome that contributes to invasion and treatment resistance. For example, reactive astrocytes that are induced by the JAK/STAT pathway could be targeted by a JAK inhibitor such as ruxolitinib, a drug that was initially developed for myelofibrosis.

The broader immune system can modify the therapeutic response in GBM. For example, the gut–brain axis has recently been associated with neurological disorders [161]. Studies of GBM have identified different compositions of the gut microbiome in both patients and GBM-bearing mice compared to healthy controls [162]. There is also an association between antibiotic treatment and GBM growth in mice [163], and a recent study showed that GBM can present bacterial epitopes that drive a specific T cell response [164]. As observed in other tumour types, the gut microbiome composition can alter ICI efficacy [165,166]. Therefore, further evaluation of the gut–brain axis is warranted to optimise therapeutic strategies using GBM immunotherapy.

In conclusion, recent studies and technological developments have advanced our understanding of the complex and intricate interactions between glioblastoma and the tumour microenvironment, providing insights into how glioblastoma evades current treatments and paving the way towards more effective therapies in the future.

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
