# Peer review of "Therapeutic Targeting of Glioblastoma and the Interactions with Its Microenvironment"

_cancers, 2023, doi:10.3390/cancers15245790_

Round 1

Reviewer 1 Report

Comments and Suggestions for Authors

Title: Therapeutic targeting of glioblastoma and the interactions of its microenvironment.

Authors: Vassilis Genoud, Ben Kinnersley, Nicholas F. Brown, Diego Ottaviani and Paul Mulholland

Comments

Overall, this is an outstanding review article written by this group.   Although the manuscript is very well written, the authors should include a section on the problem of resistance to therapy emphasizing the key determinants of drug resistance, including tumor burden, growth kinetics, heterogeneity, the immune system, and the microenvironment; undruggable cancer drivers; and the many consequences of applying therapeutic pressures.

I have no hesitation to recommend the editors to consider this review article for publication.

Author Response

Dear Editors and Reviewers,

We deeply appreciate the comprehensive feedback on our review titled " Therapeutic targeting of glioblastoma and the interactions with its microenvironment ". We have carefully reviewed all comments and suggestions, and below, we present a detailed point-by-point response.

Reviewer 1:

the authors should include a section on the problem of resistance to therapy emphasizing the key determinants of drug resistance, including tumor burden, growth kinetics, heterogeneity, the immune system, and the microenvironment; undruggable cancer drivers; and the many consequences of applying therapeutic pressures.

RESPONSE: We thank the reviewer for their valuable insight. To address therapy resistance mechanisms without overloading the main text, we added a potential therapeutic limitations column to Table 1.

Reviewer 2 Report

Comments and Suggestions for Authors

The authors of the Review entitled "Therapeutic targeting of glioblastoma and the interactions of its microenvironment" examined the interaction between glioblastoma (GBM) and tumor microenvironment (TME).  The main goal of the Review was to analyze the molecular and cellular mechanisms of TME to define and discuss proper therapeutic strategies.

The authors make an extensive introduction on the main innovations regarding the molecular characterization of GBM. Subsequently, the work is dedicated to the analysis of individual populations in interaction with the GBM. The work is interesting and would offer important insights and insights for research on GBM. There are some imperfections and I would have expected to have more details about interactions of different cell types and GBM (i.e., neurons - microglia – myeloid cells vs GBM) rather than describing the interactions between individual cell populations and GBM.

1.       The description of some concepts is approximate and vague. The grammatical structure of the sentence is also not always correct. For this reason, some parts need to be reformulated:

Below are reported some examples:

“Glioblastoma (GBM) is a disease of significant unmet need […]”

Sentences in lines 33-35.

“Tumour treating fields (TTF), an approach that uses alternating electrical fields, was reported to confer a modest OS benefit in newly diagnosed patients with GBM but has not been widely adopted.”

“MES is not observed in normal neural development but is induced in response to injury.”

“Sentences in lines 96-101”

“Sentences in lines 176-178” 

2.       Lines 137 -140: The physiological functions of the cell populations of the central nervous system are unnecessary.

 3.       Some concepts are inserted into the text without an appropriate introduction:

Below are reported some examples:

 Lines 360-362: “Targeting TAM reprogramming through CSF-1R by depleting TAM or limiting immunosuppressive polarisation is another strategy”. Explain better. It is assumed that the reader is aware of the role of CSF-1R.

 The same is in lines 368-369: “When CD47 interacts with SIRP on TAM, it inhibits their phagocytic capabilities”. Role of CD47 and SIRP⍺ ?

 4.       A weak point of the work is represented by the figures. The figures show different cells but do not add any interesting information relating to the molecular and cellular processes described in the Review.

 5.       The authors emphasize the role of the interaction between TME and GBM. In this regard, I suggest two works that concern the involvement of the TME in the progression of GBM.

 A first work considers the interconnection of different hallmarks of the GBM and therefore the mutual interaction between the different cell populations. (Torrisi F. et al. The Hallmarks of Glioblastoma: Heterogeneity, Intercellular Crosstalk and Molecular Signature of Invasiveness and Progression. Biomedicines. 2022 Mar 30;10(4):806. doi: 10.3390/biomedicines10040806. PMID: 35453557; PMCID: PMC9031586.)

Another work analyzes the role of the immune component, through an epigenetic intervention, in determining the progression of GBM. (Gangoso E, et al. Glioblastomas acquire myeloid-affiliated transcriptional programs via epigenetic immunoediting to elicit immune evasion. Cell. 2021 Apr 29;184(9):2454-2470.e26. doi: 10.1016/j.cell.2021.03.023. Epub 2021 Apr 14. PMID: 33857425; PMCID: PMC8099351.)

 6.       Minor: make sure the acronyms are inserted correctly in the text.

 7.       Line 343: typo “REF!”

Comments on the Quality of English Language

The grammatical construction of some sentences can be improved to make reading more fluent

Author Response

Dear Editors and Reviewers,

We deeply appreciate the comprehensive feedback on our review titled " Therapeutic targeting of glioblastoma and the interactions with its microenvironment ". We have carefully reviewed all comments and suggestions, and below, we present a detailed point-by-point response.

Reviewer 2:

There are some imperfections and I would have expected to have more details about interactions of different cell types and GBM (i.e., neurons - microglia – myeloid cells vs GBM) rather than describing the interactions between individual cell populations and GBM.

  1. The description of some concepts is approximate and vague. The grammatical structure of the sentence is also not always correct. For this reason, some parts need to be reformulated:

Below are reported some examples:

“Glioblastoma (GBM) is a disease of significant unmet need […]”

Sentences in lines 33-35.

“Tumour treating fields (TTF), an approach that uses alternating electrical fields, was reported to confer a modest OS benefit in newly diagnosed patients with GBM but has not been widely adopted.”

“MES is not observed in normal neural development but is induced in response to injury.”

“Sentences in lines 96-101”

“Sentences in lines 176-178”

RESPONSE: We thank the reviewer for these constructive comments and edited a large proportion of the text for better clarity.

  1. Lines 137 -140: The physiological functions of the cell populations of the central nervous system are unnecessary.

RESPONSE: We removed this part.

  1. Some concepts are inserted into the text without an appropriate introduction:

Below are reported some examples:

Lines 360-362: “Targeting TAM reprogramming through CSF-1R by depleting TAM or limiting immunosuppressive polarisation is another strategy”. Explain better. It is assumed that the reader is aware of the role of CSF-1R.          

The same is in lines 368-369: “When CD47 interacts with SIRP⍺ on TAM, it inhibits their phagocytic capabilities”. Role of CD47 and SIRP⍺ ?   

RESPONSE: We carefully reviewed our manuscript and added better introductions for these concepts.

  1. A weak point of the work is represented by the figures. The figures show different cells but do not add any interesting information relating to the molecular and cellular processes described in the Review.

RESPONSE: The figure was redesigned to add panels related to specific mechanisms described in the text. Figure 2, with the major differences between single-cell and spatial transcriptomics, was incorporated for better clarity as Figure 1 panel E and F.

  1. The authors emphasize the role of the interaction between TME and GBM. In this regard, I suggest two works that concern the involvement of the TME in the progression of GBM.

A first work considers the interconnection of different hallmarks of the GBM and therefore the mutual interaction between the different cell populations. (Torrisi F. et al. The Hallmarks of Glioblastoma: Heterogeneity, Intercellular Crosstalk and Molecular Signature of Invasiveness and Progression. Biomedicines. 2022 Mar 30;10(4):806. doi: 10.3390/biomedicines10040806. PMID: 35453557; PMCID: PMC9031586.)

Another work analyzes the role of the immune component, through an epigenetic intervention, in determining the progression of GBM. (Gangoso E, et al. Glioblastomas acquire myeloid-affiliated transcriptional programs via epigenetic immunoediting to elicit immune evasion. Cell. 2021 Apr 29;184(9):2454-2470.e26. doi: 10.1016/j.cell.2021.03.023. Epub 2021 Apr 14. PMID: 33857425; PMCID: PMC8099351.)

RESPONSE: We thank the reviewer for this constructive comment. We went through the mentioned works and included some of their concepts in our manuscript.

  1. Minor: make sure the acronyms are inserted correctly in the text.

RESPONSE: Corrected.

  1. Line 343: typo “REF!”

RESPONSE: Corrected.

Reviewer 3 Report

Comments and Suggestions for Authors

Genoud et al. described therapeutic targeting of glioblastoma and the interactions of its microenvironment. They reviewed the network of cells within the tumor microenvironment, including glioblastoma cells, astrocytes, neurons, myeloid cells, and lymphoid cells. They suggested potential treatment approaches to simultaneously target GBM cells, the surrounding immune cells, and the cross-talk between them. I highly appreciate the well-prepared review, but there are several concerns that should be addressed:

  1. 1. A new subtitle, "Immunotherapies in Glioblastoma: Unleashing the Body's Defenses," should be provided.
  2. 2. The significant challenges and limitations of therapeutic approaches should be addressed.
  3. 3. The complex microenvironment of glioblastoma should also be provided in schematic figures.
  4. 4. There are several typos and grammar mistakes in the review that should be revised carefully.
Comments on the Quality of English Language

Moderate editing of English language required

Author Response

Dear Editors and Reviewers,

We deeply appreciate the comprehensive feedback on our review titled " Therapeutic targeting of glioblastoma and the interactions with its microenvironment ". We have carefully reviewed all comments and suggestions, and below, we present a detailed point-by-point response.

Reviewer 3:

  1. A new subtitle, "Immunotherapies in Glioblastoma: Unleashing the Body's Defenses," should be provided.
  2. The significant challenges and limitations of therapeutic approaches should be addressed.
  3. The complex microenvironment of glioblastoma should also be provided in schematic figures.
  4. There are several typos and grammar mistakes in the review that should be revised carefully.

RESPONSE: We thank the reviewer for their valuable comments. To keep the text aligned with the current layout, we believed that a new subtitle would not necessarily fit with the current structure. We added potential limitations to therapeutic approaches in table 1 and improved the figure to add more complexity and better reflect the main text. Additionally, we carefully reviewed the text for grammar and clarity issues.

Round 2

Reviewer 2 Report

Comments and Suggestions for Authors

The authors of the Review have improved the quality of the work.

The figures provide additional information that helps the reader to link with the text. Figure 1C clarifies the immune cell populations.

The idea of ​​UMAP plot indicating the diversity of cell populations in GBM is highly appreciated for the work.

In this regard, I would only ask the authors to add some details and descriptions on this part in the text.

Comments on the Quality of English Language

I suggest to carry out a final check to simplify some sentences and make reading easier.

Author Response

Dear Editors and Reviewers,

We deeply appreciate the comprehensive feedback on our review titled " Therapeutic targeting of glioblastoma and the interactions with its microenvironment ". We have carefully reviewed the last suggestion and have addressed it in the new manuscript. We added a paragraph introducing specifically Figure 1E and F at the beginning of section 6. Additionally, we reviewed the whole manuscript to improve clarity.

Reviewer 3 Report

Comments and Suggestions for Authors

Authors have addressed my concerns. I recommend the acceptance of the paper.

Author Response

Dear Editors and Reviewers,

We deeply appreciate the comprehensive feedback on our review titled " Therapeutic targeting of glioblastoma and the interactions with its microenvironment ".